# Sebum Components Dampen the Efficacy of Skin Disinfectants against *Cutibacterium acnes* Biofilms

**DOI:** 10.3390/microorganisms12020271

**Published:** 2024-01-27

**Authors:** Dilini Kumaran, Sandra Ramirez-Arcos

**Affiliations:** 1Innovation & Portfolio Management, Canadian Blood Services, Ottawa, ON K1G 4J5, Canada; dilini.kumaran@blood.ca; 2Department of Biochemistry, Microbiology and Immunology, University of Ottawa, Ottawa, ON K1N 6N5, Canada

**Keywords:** blood donor disinfectant, *Cutibacterium acnes* biofilms, disinfection efficacy, skin sebum

## Abstract

At Canadian Blood Services, despite the use of 2% chlorhexidine and 70% isopropyl alcohol (standard disinfectant, SD) prior to venipuncture, *Cutibacterium acnes* evades eradication and is a major contaminant of platelet concentrates (PCs). Since *C. acnes* forms bacterial aggregates known as biofilms in the sebaceous niches of the skin, this study aimed to assess whether sebum-like components impact disinfectant efficacy against *C. acnes* leading to its dominance as a PC contaminant. *C. acnes* mono-species and dual-species biofilms (*C. acnes*s and a transfusion-relevant *Staphylococcus aureus* isolate) were formed in the presence and absence of sebum-like components and exposed to SD, a hypochlorous acid-based disinfectant (Clinisept+, CP), or a combination of both disinfectants to assess disinfectant efficacy. Our data indicate that sebum-like components significantly reduce the disinfectant efficacy of all disinfectant strategies tested against *C. acnes* in both biofilm models. Furthermore, though none of the disinfectants led to bacterial eradication, the susceptibility of *C. acnes* to disinfectants was heightened in an isolate-dependent manner when grown in the presence of *S. aureus*. The reduction of skin disinfection efficacy in the presence of sebum may contribute to the overrepresentation of *C. acnes* as a PC contaminant and highlights the need for improved disinfection strategies.

## 1. Introduction

Human skin is rich in microbial diversity, and the composition of these bacterial populations can vary by gender [1], site on the skin [2], gut health [3], and environmental factors [4]. Multiple studies have demonstrated that bacterial skin isolates can form aggregates called biofilms [5,6]. The ability to form biofilms is an important virulence factor conferring to bacteria that dwell within these communities, a heightened ability to resist the action of antimicrobials and the ability to evade host immune responses [7]. It is therefore unsurprising that biofilms have been associated with disease states on the skin [8] and in recalcitrant, deep seated chronic infections such as those encountered in periprosthetic joint infections [9], urinary tract infections [10], and in the lungs of cystic fibrosis patients [11] to name a few. Consequently, skin disinfection prior to invasive procedures, is an important strategy to prevent infection by diminishing the risk of the inadvertent introduction of bacteria from the surface of the skin into the patient.

*Cutibacterium acnes* is an anaerobic aerotolerant member of the skin flora and is one of the most represented bacterial species on the skin [2]. *C. acnes* thrives in regions rich in sebum where it can form biofilms that can extend into the deep layers of the terminal hair follicle [6]. Generally considered a harmless commensal, *C. acnes* has often been dismissed as a contaminant when isolated from clinical samples [12] However, in recent years, there has been a growing awareness of the association of this bacterium with chronic slow developing infections such as infectious endocarditis [13], bone infections [14], and prosthetic device infections [9].

Canadian Blood Servies manufactures platelet concentrates (PCs), a blood product used to treat patients with platelet deficiencies. PCs consist of platelets suspended in 100% plasma or in plasma diluted in platelet additive solution and are stored in gas permeable plastic containers at 20–24 °C under agitation to maintain platelet functionality. However, these storage conditions can promote the proliferation of bacterial contaminants introduced during blood collection. To mitigate the risk of transfusing contaminated PCs, this blood product is routinely screened for the presence of bacteria with the automated BACT/ALERT 3D culture system (bioMérieux, Montreal, QC, Canada) [15]. Hemovigilance studies at Canadian Blood Services have revealed that *C. acnes* is the most isolated bacterial contaminant of PCs accounting for ~70% of all bacteria isolated from positive cultures [15]. Due to the slow growing nature of *C. acnes* in vitro, it is often detected after contaminated units are transfused into vulnerable patients [15]. This occurs despite the regimented use of an aqueous disinfectant solution (chlorhexidine gluconate (2%, *v*/*v*) and isopropyl alcohol (70%, *v*/*v*)) to disinfect the donor’s skin (30 s) prior to venipuncture [16]. Though only a handful of mild transfusion reactions have been attributed to *C. acnes* contaminated blood products [17,18], it is difficult to ascertain the long-term impacts of these transfusion events.

This study aimed to investigate factors that contribute to the over representation of *C. acnes* as a PC contaminant, by assessing the impact that sebum-like components have on biofilm formation and disinfectant efficacy in mono-species and dual-species biofilm models established with *C. acnes* isolates derived from PCs and a transfusion-relevant *Staphylococcus aureus* isolate.

## 2. Materials and Methods

### 2.1. Bacterial Strains

Twenty *C. acnes* isolates obtained from PCs that were transfused to patients originating from different geographical regions of Canada (2018–2019) were assessed in this study and have been listed in Table 1. The ATCC 6919 *C. acnes* strain was used as a control for multiplex PCR and for preliminary biofilm assays, and a transfusion-relevant *S. aureus* isolate (CBS 2016-05) [19] was used for dual biofilm assays.

### 2.2. Media and Reagents

All bacterial isolates were cultured on tryptic soy agar supplemented with 5% sheep blood (blood agar, BA, Fisher Scientific, Hampton, NH, USA) unless stated otherwise. Bacterial suspensions for biofilm assays were prepared in Brain Heart Infusion media (BHI). Sebum-like components including triolein, squalene, and olive oil (source of oleic acid) were purchased from Sigma Aldrich (Oakville, ON, Canada) while pure jojoba oil was sourced from Now Solutions (Toronto, ON, Canada). These components were either used individually or in combination in previously described ratios [20] to produce a sebum-like emulsion referred to as “SL” in biofilm assays. A neutralizing solution consisting of 10% (*v*/*v*) Tween 80 (Sigma Aldrich, Oakville, ON, Canada), 3% (m/v) lecithin (Fisher Scientific, Ottawa, ON, Canada) and 0.3% (m/v) sodium thiosulfate (Sigma Aldrich, Oakville, ON, Canada) was assessed as per ASTM E1054-08 (Standard Test Methods for Evaluation of Inactivators of Antimicrobial Agents) [21] and employed in disinfectant efficacy assays. A 20% stock solution of chlorhexidine gluconate (Sigma Aldrich, Oakville, ON, Canada) and pure isopropyl alcohol (Fisher Scientific, Ottawa, ON, Canada) were diluted in sterile water to prepare an aqueous disinfectant solution consisting of 2% chlorhexidine gluconate and 70% isopropyl alcohol (standard disinfectant, SD) and used immediately in disinfectant efficacy assays. Clinisept + (CP) (Clinical Health Technologies, Peterborough, UK) was used as an alternate disinfectant in this study. 

### 2.3. Phylotyping C. acnes PC Isolates

*C. acnes* PC isolates were sub-cultured on BA and incubated anaerobically at 37 °C for 72 h. Bacterial colonies were collected using a sterile cotton swab and resuspended in Tris EDTA buffer (pH 7.4) and heated to 80 °C for 15 min. The bacterial suspension was then treated with 1 × 10^6^ units of lysozyme and RNase A (final concentration 100 µg/mL) and incubated at 37 °C for 2 h. Proteinase K (final concentration of 0.1 µg/µL) and 6% SDS solution at 0.1× of the total volume were added and incubated at 37 °C with agitation (160 rpm) for 2 h. The samples were then placed at 55 °C for 30 min, and genomic DNA was extracted using an equal volume of chloroform: phenol: isoamyl alcohol (25:24:1) and precipitated with sodium acetate and ethanol. A touchdown-multiplex PCR was performed, and the isolates were categorized into different phylotypes based on the banding patterns obtained on a 1.2% agarose gel as previously described [22].

### 2.4. Preliminary Assessment of C. acnes Biofilm Formation with Oleic Acid Supplementation

The twenty *C. acnes* PC isolates and the ATCC 6919 isolate that served as a positive control were sub-cultured on BA for 72 h at 37 °C under anaerobic conditions. Bacterial suspensions corresponding to ~10^7^ colony forming units (CFU)/mL were prepared in BHI using a Densimat machine (reading of ~0.15) (bioMérieux, Montreal, QC, Canada). This bacterial suspension was used to seed the wells of 48-well plates that were either uncoated or coated with olive oil. Coated and uncoated wells seeded with media in the absence of bacteria served as the background control. The plates were incubated statically under anaerobic conditions for 7 days at 37 °C. Planktonic cells were removed, adhered cells were washed with sterile water, and the biofilms were fixed for 30 min with 100% methanol. The fixed biofilms were stained with a crystal violet solution for 30 min, following which biofilms were washed to remove excess stain. An aqueous solution of 20% (*v*/*v*) methanol and 5% (*v*/*v*) acetic acid was used to solubilize the adhered stain, which was quantified using a spectrophotometer at 492 nm. Each condition was tested in technical duplicates and in biological triplicates. 

### 2.5. Impact of Sebum-like Components on C. acnes Biofilm Formation

Four *C. acnes* PC isolates (BPNBT-19269, 19195, 19329, and 19227) representing the different phylotypes identified were chosen for successive experiments based on preliminary biofilm data. Biofilm formation was assessed as described above; however, wells either remained uncoated (control), or were coated with individual sebum-like components (triolein, squalene, jojoba oil, and olive oil) or SL. Each condition was tested in technical duplicates and biological triplicates.

### 2.6. Confocal Microscopy

*C. acnes* biofilms of the four representative PC isolates were established over 7 days at 37 °C under anaerobic conditions in 8-chambered permanox microscope slides (Thermofisher) coated with SL. Biofilms formed in uncoated chambers served as a control. Biofilms were washed and fixed with 2.5% glutaraldehyde for 2 h at room temperature. The fixative was then removed, biofilms washed with phosphate-buffered saline (PBS), and Fluoroshield (Sigma Aldrich, Oakville, ON, Canada) a mounting medium containing DAPI was added to each chamber and was visualized using the LSM 800 microscope and Zeiss EC Plan-Neofluar 10×/0.3 objective lens with excitation by a 405 nm solid state laser.

### 2.7. C. acnes and S. aureus Dual-Species Biofilm Formation in the Presence of Sebum-like Components

The wells of 48-well plates were coated with SL and were seeded with the bacterial suspensions of the four representative *C. acnes* PC isolates (BPNTBT-19269, 19195, 19329, and 19227). The plates were incubated statically for 6 days at 37 °C under anaerobic conditions, at which point 0.25 mL of the planktonic cells were removed and replaced with 0.25 mL of *S. aureus* (CBS 2016-05) bacterial suspension prepared in BHI (10^7^ CFU/mL). The plates were re-incubated statically for an additional 24 h under anaerobic conditions at 37 °C. Wells containing mono-species biofilms served as biofilm controls, while wells containing media alone served as background control. Once biofilms were established, the biofilms were either stained as described above or enumerated by dislodging the biofilms and plating serial dilutions on BA plates to determine bacterial counts of each species (anaerobic incubation, 37 °C, up to 96 h). Each condition was tested using technical duplicates and in biological triplicates.

### 2.8. Disinfectant Efficacy Assays

#### 2.8.1. Mono-Species Biofilms

Washed preformed biofilms of BPNTBT-19269, 19195, 19329, and 19227 established in either individual sebum-like components or SL coated wells or uncoated control wells were exposed to either 0.5 mL of (a) SD for 30 s as is performed on the skin during donor skin disinfection prior to venipuncture, (b) CP for 2 min as recommended by the manufacturer, (c) to the two disinfectants successively with exposure to SD first followed by CP or (d) to CP first followed by SD. After the appropriate contact time had elapsed, the disinfectant was removed, 0.5 mL of neutralizing solution was immediately added per well, and the biofilm was dislodged. Bacterial suspensions of duplicate wells were immediately pooled and added to 4 mL of neutralizing solution (total 5 mL) and vortexed. The suspension was incubated at 20–24 °C for 5 min to ensure complete neutralization of residual disinfectant. The suspension was vortexed vigorously, serially diluted, plated on BA plates and incubated anaerobically at 37 °C for 96 h. Biofilms not exposed to the disinfectant provided baseline bacterial counts and served as a control. Each condition was tested in biological triplicates.

#### 2.8.2. Disinfectant Efficacy against Dual-Species Biofilms

The efficacy of SD and CP was assessed on pre-washed mono-species (*C. acnes* or *S. aureus*) and dual-species (*C. acnes* and *S. aureus*) biofilms established in wells coated with SL by exposing biofilms to either SD, CP or successively to SD followed by CP as described above. Biofilms not exposed to the disinfectant provided baseline bacterial counts and served as a control. Each condition was tested using technical duplicates and in biological triplicates.

### 2.9. Biofilm Matrix Disruption Assay

Biofilms of the four *C. acnes* isolates were established in 6-well plates that were either uncoated or coated with SL as described above with the bacterial seed volume modified to 3 mL. Biofilms were washed with PBS, and the test wells were treated with either proteinase K (100 µg/mL) in 20 mM Tris buffer (pH 7.5) and 100 mM NaCl, DNAse 1 in PBS (30 ng/mL), or 10 mM sodium metaperiodate in 50 mM sodium acetate buffer pH 4.5. Biofilms formed in coated and uncoated wells treated with the diluent buffers alone served as a control. Plates were incubated at 37 °C for 1 h, after which the supernatant was removed, and biofilms were fixed, stained, and quantified as described above. The percent composition of each component was obtained by subtracting the readings obtained for test wells from the control wells and dividing that value by the test well readings. Each condition was tested in biological triplicates.

### 2.10. Statistical Analyses

Two-way ANOVA analysis was used to compare the differences in disinfectant efficacy against *C. acnes* biofilms established in different sebum-like components. Statistical differences in experiments comparing coated and uncoated wells were performed using either one-way ANOVA or *t*-test (Student and Welch) as indicated.

## 3. Results

### 3.1. C. acnes PC Isolates Predominantly Belong to Phylotypes IB and II 

Multiplex PCR indicated that 50% of the 20 PC isolates tested belong to phylotype II, followed by phylotype IB (30%), while phylotypes III and IA accounted for 15% and 5% of the isolates tested, respectively (Table 2). Control strain ATCC 6919 isolate was categorized as phylotype IA.

### 3.2. The Addition of Olive Oil as a Source of Oleic Acid Does Not Enhance the Biofilm Formation of all C. acnes Isolates

All *C. acnes* isolates tested were able to form biofilms in both olive oil coated and uncoated wells (Figure 1). Isolates BPNBT-19269, BPNBT-19295, and BPNBT-19352 formed significantly more biofilms in olive oil coated wells (*p* ≤ 0.05), while isolates BPNBT-19119, BPNBT-19005, BPNBT-19054, BPNBT-19354, and BPNBT-19355 formed significantly more biofilms in uncoated wells (*p* ≤ 0.05). All other PC isolates tested exhibited similar biofilm formation in both conditions (Figure 1). Four *C. acnes* isolates were chosen for successive experiments representing the four *C. acnes* phylotypes identified (Table 2), three of these isolates BPNBT-19195 (phylotype IB), 19329 (phylotype II), and 19227 (phylotype III) display similar biofilm formation in coated and uncoated wells, while isolate BPNBT-19269 was chosen since it was the only PC isolate tested that was categorized as phylotype IA.

### 3.3. C. acnes Biofilm Formation Is Impacted by Different Sebum-like Components in an Isolate-Dependent Manner

The biofilm formation of the four *C. acnes* PC isolates was generally unchanged in the presence of sebum-like components compared to the uncoated controls (Figure 2). Isolate BPNBT-19269 displayed a significant increase in biofilm formation in wells coated with triolein and SL (*p* ≤ 0.05) compared to control uncoated wells. It should be noted; however, that the biofilms formed in uncoated wells were more easily dislodged during the staining and de-staining process and therefore observed reductions reflect the loss during processing. Squalene on the other hand significantly reduced the biofilm formation of isolate BPNBT-19195 compared to the uncoated control; however, no differences were observed in biofilm formation in the presence of all the other coatings tested. Significantly higher biofilm formation was observed for isolate BPNBT-19227 in wells coated with triolein (*p* ≤ 0.05) compared to uncoated wells, with similar biofilm formation observed in all other coated wells. Interestingly, no significant differences in biofilm formation between coated and uncoated wells were observed for isolate BPNBT-19329 (Figure 2). Since SL reflects the pilosebaceous niche more closely, successive experiments were performed using this coating. 

### 3.4. Denser C. acnes Biofilms Are Formed in the Presence of Sebum-like Components

The confocal microscopy images of the biofilms produced in wells coated with SL indicate that the biofilms are denser (visualized by increased fluorescence intensity) than biofilms established in uncoated wells (Figure 3A). Biofilms produced by three of the four isolates BPNTBT-19195, 19329, and 19227 appear to be denser and uniform in wells coated with SL, except for isolate BPNBT-19269 which appears to produce irregular aggregates across the biofilm forming surface consistent with the rough appearance of these biofilms visible to the naked eye. Total bacterial counts obtained for biofilms established in 48-well plates in coated and uncoated wells indicate slight increases in biofilm load (Figure 3B) in wells coated with SL.

### 3.5. The Presence of S. aureus in Dual-Species Biofilms Affect C. acnes Growth in a Strain-Dependent Manner

The readings obtained from the crystal violet biofilm assays for the dual-species biofilms of *C. acnes* isolates BPNBT-19269 and 19329 with *S. aureus* CBS2016 did not differ significantly from the combined readings of the mono-species biofilms obtained for the two isolates (Figure 4A). However, the readings of the dual-species biofilms formed with BPNBT-19195 and 19227 with *S. aureus* were significantly higher (*p* ≤ 0.05) than the combined readings of the mono-species biofilms (Figure 4A). A comparison of the bacterial concentration in dual-species and mono-species biofilms indicates that three of the four *C. acnes* isolates (BPNBT-19269, 19329, and 19227) have a significantly higher bacterial concentration in the dual-species model (*p* ≤ 0.05) when compared to bacterial concentrations obtained in the mono-species model (Figure 4B). On the other hand, *S. aureus* concentrations in the dual-species model were significantly elevated (*p* ≤ 0.05) in only two (BPNBT-19195, and 19329) of the four isolates tested, while a significant reduction (*p* ≤ 0.05) in the *S. aureus* bacterial load was observed when grown in the biofilm model with the *C. acnes* BPNBT-19269 isolate (Figure 4C).

### 3.6. Sebum-like Components Negatively Impact the Efficacy of the Standard Donor Skin Disinfectant and a Hypochlorous Acid-Based Skin Disinfectant

#### 3.6.1. Mono-Species *C. acnes* Biofilms Resist Disinfection in the Presence of Sebum-like Components

The total bacterial counts obtained from the disinfectant efficacy assays performed on all four *C. acnes* mono-species biofilms displayed a significant reduction (*p* ≤ 0.01) in the efficacy of SD against biofilms established in wells coated with individual sebum-like components and SL when compared to uncoated wells (Figure 5A). A comparison of the resistance to disinfection displayed by biofilms formed in the presence of the different sebum-like components indicated that there were no significant differences in the inhibition of disinfectant efficacy as a result of the coating. Further assessment of the efficacy of SD and CP used individually on *C. acnes* biofilms established in wells coated with SL indicates that the efficacy of both disinfectants was reduced significantly (*p* ≤ 0.05) (Figure 5B). Furthermore, using a combination of the two disinfectants did not increase the disinfectant efficacy against biofilms formed in coated wells to that observed in uncoated wells, and was significantly lower than the efficacy observed in uncoated wells (Figure 5B). A comparison of the efficacy of the two disinfectants used successively indicated that there was no significant difference in the efficacy irrespective of the order in which the disinfectants were applied to the biofilms prepared in wells coated with SL (Figure 5B).

#### 3.6.2. *S. aureus* in Mono- and Dual-Species Biofilm Models Resist Elimination by Skin Disinfectants in the Presence of Sebum-like Components

*S. aureus* was not eliminated in either mono-species or dual-species biofilms when SD, CP, or a combination of the two disinfectants was used (Figure 6). A significant increase (*p* ≤ 0.05) in CP disinfectant efficacy was observed against *S. aureus* and *C. acnes* BPNBT-19269 (Figure 6A) in the dual-species biofilm model compared to the respective efficacies observed in the mono-species models. A similar trend was observed for *C. acnes* BPNBT-19195 in the dual-species biofilm model (Figure 6B). The disinfectant efficacy of the three treatments against isolate BPNBT-19329 was significantly increased (*p* ≤ 0.05) in the dual-species biofilm model when compared to the mono-species biofilm model; however, an increase in disinfectant efficacy against the *S. aureus* isolate in the dual-species model was only observed when a combination of SD and CP was used (Figure 6C). A significant reduction in disinfectant efficacy against *S. aureus* in the dual-species model with *C. acnes* BPNBT-19227 was only observed when treated with SD (Figure 6D).

### 3.7. Sebum-like Components Impact the Composition of C. acnes Biofilm Matrix in a Strain-Dependent Manner

In all four *C. acnes* isolates biofilms tested, the extracellular matrix was found to be mainly composed of protein and extracellular DNA in both uncoated and SL coated wells (Figure 7). The polysaccharide composition of biofilm matrices was almost negligible in BPNBT-19269, 19195, and 19227, and was low in isolate BPNBT-19329 (8–18%). There was a significant reduction in the protein and eDNA composition of the extracellular biofilm matrix of two of the four *C. acnes* isolates (BPNBT-19269, BPNBT-19195) established in SL coated wells compared to the uncoated control, accompanied by no changes in the polysaccharide composition. In isolates BPNBT-19329 and BPNBT-19227, no significant changes in biofilm matrix composition were observed when biofilms established in SL coated wells were compared to uncoated wells (Figure 7).

## 4. Discussion

The disinfection of donor skin prior to blood collection serves as a key intervention to prevent bacterial contamination of blood products. Disinfectant solutions consisting of chlorhexidine gluconate and isopropyl alcohol are considered the gold standard of skin disinfectants [23] and are used at Canadian Blood Services and by other blood suppliers [16,24,25]. However, bacterial contamination of blood products continues to occur, highlighting the limitations of skin disinfection [15,17,18]. The lipophilic bacterium *C. acnes* thrives in the sebaceous niches of human skin as biofilms [6] and is the predominant species isolated during PC screening with culture methods [15,17]. This is despite the fact that the antecubital region, the site of blood donor venipuncture, is mostly populated with *Corynebacterium* spp. and coagulase negative staphylococcal species [2]. In this study, the efficacy of the standard disinfectant (chlorhexidine and isopropyl alcohol) and an alternate hypochlorous acid-based disinfectant (Clinisept+) was tested against *C. acnes* biofilms in the presence and absence sebum-like components in mono- and dual-species models. Our data demonstrate that the presence of sebum-like components can significantly reduce the disinfectant efficacy of both disinfectants tested. The dampened efficacy of the standard disinfectant coupled with the poor penetration properties of chlorhexidine gluconate in alcohol [26], may contribute to the ability of *C. acnes* to escape donor skin disinfection resulting in this species being overrepresented as a blood product contaminant.

In this study, twenty *C. acnes* isolates obtained from contaminated PCs were assessed and 80% were found to belong to phylotypes IB and II. Notably, these phylotypes have been associated with deep seated infections which are often associated with biofilm formation in the clinical setting [27]. It is therefore important to understand the factors contributing to the ineffective disinfection of donor skin if the risk to transfusion patients is to be mitigated. Bacteria associated with biofilms display a heightened resistance to antimicrobials and the environmental conditions the biofilms are established in may impact these observed characteristics. Sebum components like squalene, oleic acid, and palmitic acid (jojoba oil) have been shown to inhibit or diminish biofilm formation and possess antimicrobial properties [28,29,30]. However, the work presented herein demonstrates that the presence of sebum-like components (individual or in combination) does not significantly impact the overall biofilm formation of *C. acnes* but can result in a slight increase in bacterial load. Studies that assess the impact of sebum components on biofilm formation of a larger numbers of *C. acnes* isolates could be performed to confirm these findings.

Of note, tween 80 (polyethylene sorbitol ester of oleic acid) and the sodium salt of oleic acid are important components of neutralizers of chlorhexidine gluconate and isopropyl alcohol [31]. Since oleic acid is an important component of sebum, it is possible that the mere presence of sebum may act as a neutralizing shield against donor skin disinfection, particularly in niches of the skin where it is found in high concentrations, leading to reduced efficacy against bacteria like *C. acnes* that reside there. The extracellular matrix (ECM) of biofilms plays various roles that impact the unique properties displayed by biofilms. In the context of antimicrobial activity, the ECM may prevent or retard the diffusion of antimicrobials or sequester these agents preventing contact with their bacterial targets [32,33]. The ECM is generally composed of proteins, extra-cellular DNA, polysaccharides, and lipids [34] and the composition of these components can vary depending on the environment in which the biofilm is formed [35]. In our study since the biofilms were established directly on lipid coating, it was not feasible to assess the ECM fraction of the biofilms without inadvertently contaminating samples with the coating, instead we aimed to assess the composition of other components to determine if any significant changes could infer the incorporation of lipid. Our data indicate that there is a significant reduction in the protein and e-DNA content of the biofilms produced by two isolates (BPNBT-19269 and 19195) in SL coated wells that is not offset with a corresponding increase in polysaccharide content. This could imply that lipids found in the environment (coating) could potentially be incorporated into the ECM. Since all isolates displayed heightened resistance to disinfection, it is not possible to elucidate if and to what extent the potential incorporation of lipid into the ECM has on the observed resistance. Additionally, changes in the lipid composition of the bacterial membrane have been reported to modulate the fluidity of the membrane in the biofilm state thereby changing the susceptibility of bacteria to antimicrobials [36]. In our study, an assessment of bacterial membrane composition was not performed due to technical difficulties in preventing the contamination of coating into the membrane sample. As a result, the incorporation of exogenous lipids from the coating into the bacterial membrane in *C. acnes* biofilms cannot be dismissed as a contributing factor for the increased resistance to disinfection observed. Future studies using a modified sebum biofilm model allowing for reduced direct contact with the coating could facilitate an in-depth analysis of both the ECM and biofilm associated bacterial membrane.

In nature, biofilms are often found in multi-species communities, and members of these communities may exert influence on each other thereby impacting growth and antimicrobial susceptibility [37,38]. Abbott et al. demonstrated that *C. acnes* belonging to phylotype IA can inhibit the growth and/or delay the maturation of *S. aureus* biofilms leading to reduced biofilm mass [37]. The data obtained in our dual-species model that included *S. aureus* and *C. acnes* isolate BPNBT-19269 (phylotype IA) was consistent with these findings since a significant reduction in *S. aureus* concentration was observed. However, this reduction was not observed when dual-species biofilms were established with *C. acnes* isolates belonging to other phylotypes, indicating that the inhibitory properties may be *C. acnes* phylotype dependent. The susceptibility of *S. aureus* to antimicrobials has been shown to vary in the presence of other species, for instance, *C. acnes* derived factors significantly enhance the sensitivity of *S. aureus* biofilms to antimicrobials [37], while dual-species biofilms consisting of *S. aureus* and *C. albicans* increased the resistance of the former to vancomycin a hundred-fold [38]. Our data indicate that *C. acnes* does not enhance the sensitivity of *S. aureus* to SD in three out of the four dual-species models assessed. However, the sensitivity of three *C. acnes* isolates to oxidant damage by CP was enhanced when grown in a dual-species biofilm with *S. aureus*. The mechanisms engaged in antioxidant responses are not fully elucidated in *C. acnes*; however, our data show that the presence of *S. aureus* disrupts these mechanisms when grown together in a dual-species biofilm model. Previous work has shown that the biosynthesis of staphlyoxanthin, an antioxidant produced by *S. aureus,* is inhibited by exposure to squalene thereby increasing its susceptibility to oxidants [39]. It is therefore plausible that in the dual-species model, *S. aureus* or its secreted factors may directly inhibit the synthesis of antioxidants or may sensitize *C. acnes* to sebum components which in turn interfere with the antioxidant response. Taken together our results indicate that though the sebum environment affords protection against disinfection, this resistance can be modulated in multi-species communities, and these variations may be phylotype dependent.

Though the in vitro model used in this study mimics a sebum-like environment, there are components of the pilosebaceous unit that are not represented. For instance, sapienic acid, a fatty acid unique to human sebum that is known to have antibacterial properties, [40] was not included. Furthermore, the cellular structures forming the different layers of skin and the complex crosstalk between bacteria and the immune system required to maintain homeostasis on the skin [41] is not reflected in our model. Nonetheless, it should be noted that the quantitative suspension test [42] and antimicrobial dilution assays [43] are the standard methods employed to assess the efficacy of disinfectants. These standard methods do not assess the impact of biofilm formation or other components that may be present in the natural site of disinfection on disinfectant efficacy. Our data indicate that the omission of these components and biofilm formation may lead to the overestimation of the efficacy of disinfectants.

## 5. Conclusions

The presence of sebum components on the skin may dampen the efficacy of the gold standard disinfectant consisting of 2% chlorhexidine and 70% isopropyl alcohol. This study highlights how the low penetration properties of this disinfectant together with the inhibitory effects of sebum components could contribute to the overrepresentation of *C. acnes* as a PC contaminant. Furthermore, this study emphasizes the need for biofilm models that better reflect the natural niche that a bacterium inhabits in order to have a more accurate understanding of how they respond to different stressors, including antimicrobials and disinfectants. Finally, understanding the shortcomings of current disinfectant strategies provides a platform on which future innovation aimed at developing more effective disinfectants that are less susceptible to the inhibitory properties of sebum can be built.

## Figures and Tables

**Figure 1 microorganisms-12-00271-f001:**
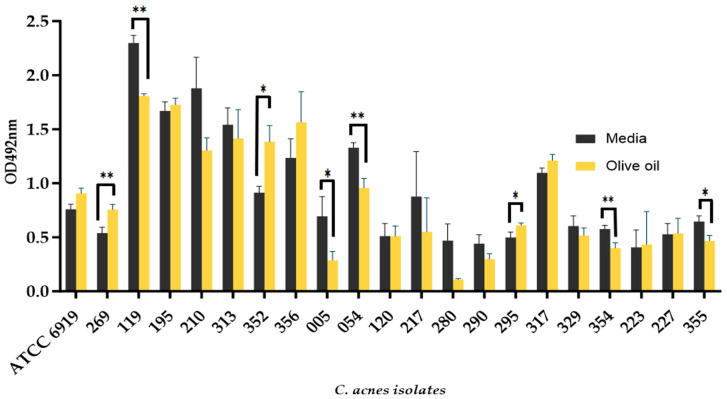
Biofilm formation of *C. acnes* PC isolates. Biofilm formation of *C. acnes* PC isolates in uncoated wells (black bars) and wells coated with olive oil a source of oleic acid (yellow bars). Student’s *t*-test analysis, *p* ≤ 0.05 (*), *p* ≤ 0.01 (**). OD= Optical Density. *n*= 3.

**Figure 2 microorganisms-12-00271-f002:**
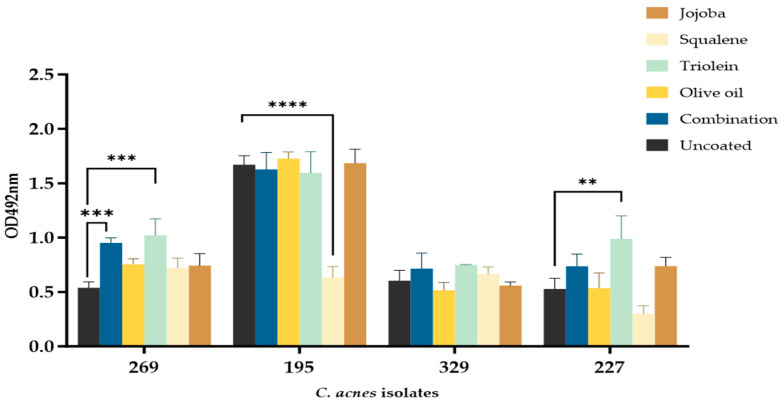
Biofilm formation of *C. acnes* PC isolates in the presence of sebum-like components. Biofilm formation of *C. acnes* PC isolates in uncoated wells and wells coated with individual sebum-like components or the combination (i.e., SL) as assessed by the crystal violet biofilm assay following incubation at 37 °C for 7 days under anaerobic conditions. *n* = 3. Significance assessed by One-way ANOVA analysis, *p* ≤ 0.01 (**), *p* ≤ 0.001 (***), and *p* ≤ 0.0001 (****).

**Figure 3 microorganisms-12-00271-f003:**
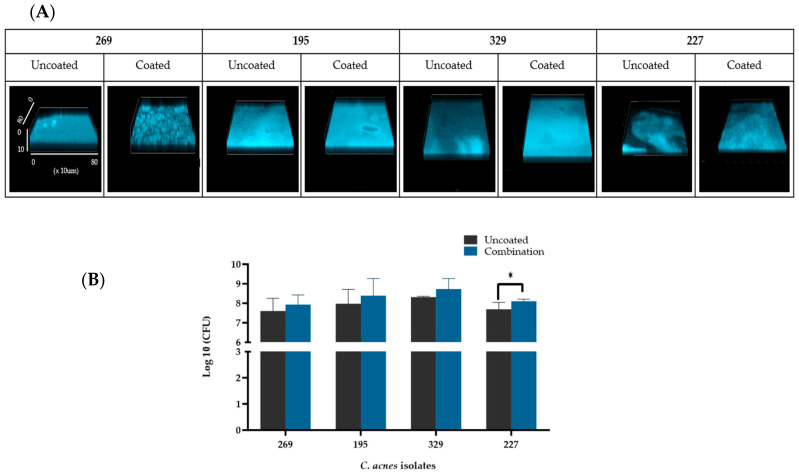
Qualitative and quantitative comparison of *C. acnes* biofilms produced in the absence and presence of sebum-like components. (**A**) Confocal microscopy images of *C. acnes* biofilms of the four PC isolates established in uncoated or wells coated with aa combination of sebum-like components (i.e., SL) for 7 days at 37 °C under anaerobic conditions stained with Fluoroshield. Fluorescence intensity if proportional to bacterial density. The scale in the first image (269 uncoated) applies to all images. (**B**) Total bacterial counts of *C. acnes* derived from biofilms established in 48-well plates in the absence and presence of a combination of sebum-like components (i.e., SL) represented as the Log10 value of colony forming units (CFU) (*n* = 6). Student’s *t*-test analysis, *p* ≤ 0.05 (*).

**Figure 4 microorganisms-12-00271-f004:**
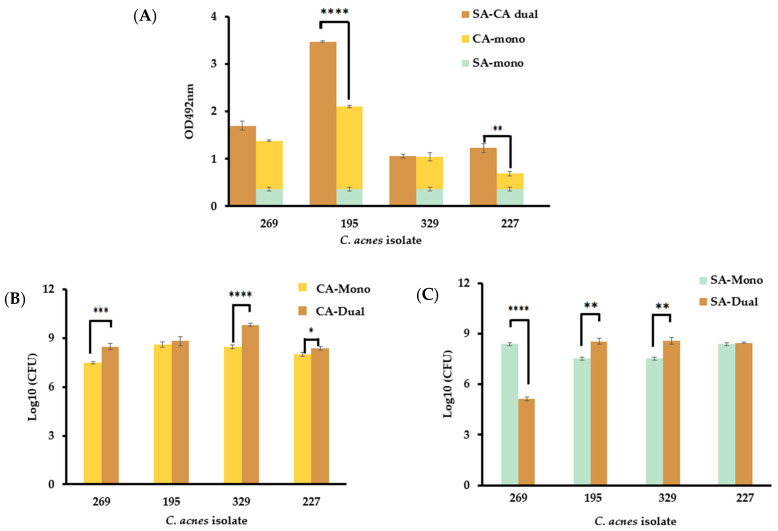
Dual-species biofilms of *S. aureus* with four PC *C. acnes* isolates. (**A**) Comparison of the optical density of *S. aureus* and *C. acnes* biofilm formation in mono- and dual-species biofilm models in the presence of SL, (**B**) bacterial load of the four *C. acnes* PC isolates in the mono- and dual-species biofilm models established in the presence of SL, (**C**) bacterial load of *S. aureus* in the mono- and dual-species biofilm models established in the presence of SL. *n* = 3, Student’s *t*-test analysis, *p* ≤ 0.05 (*), *p* ≤ 0.01 (**), *p* ≤ 0.001 (***), and *p* ≤ 0.0001 (****).

**Figure 5 microorganisms-12-00271-f005:**
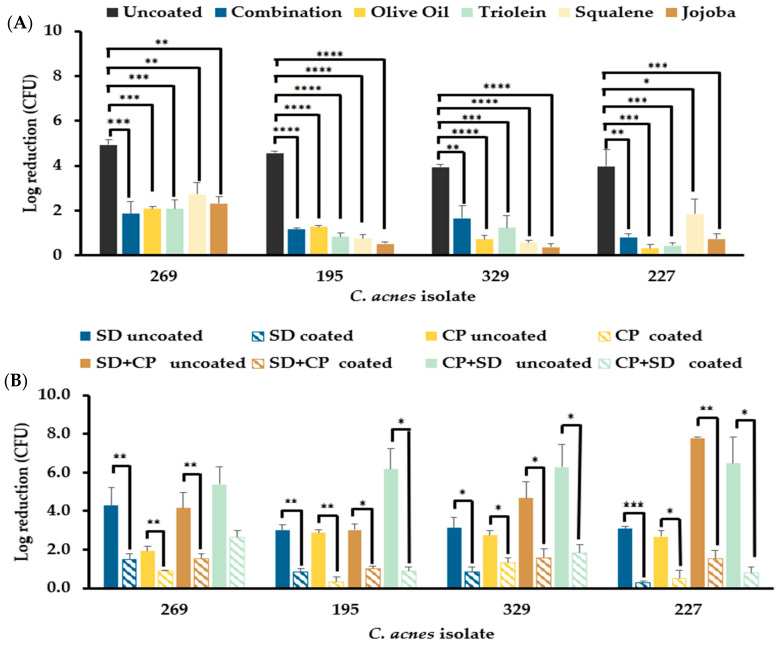
Disinfectant efficacy of skin disinfectants against *C. acnes* biofilms. (**A**) Reduction of biofilm bacterial load of the four *C. acnes* PC isolates established in the presence of individual or a combination of sebum-like components (i.e., SL) compared to the control (uncoated) following treatment with SD disinfectant (2% chlorhexidine, 70% isopropyl alcohol). (**B**) Reduction of biofilm bacterial load of biofilms established in SL compared to the control (uncoated) following treatment with either SD, Clinisept+ (CP) or successive treatment with the two disinfectants. *n* = 3, *p* ≤ 0.05 (*), statistical analysis (**A**) One-way ANOVA, (**B**) Welch’s *t*-test, *p* ≤ 0.01 (**), *p* ≤ 0.001 (***), and *p* ≤ 0.0001 (****).

**Figure 6 microorganisms-12-00271-f006:**
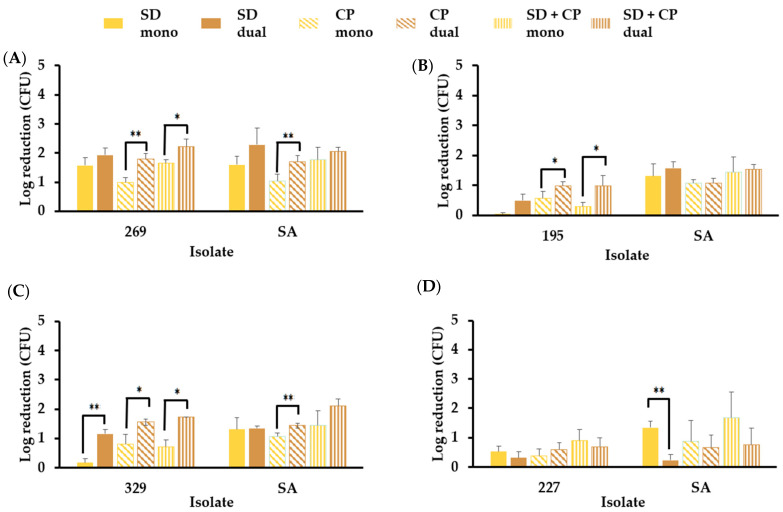
Disinfectant efficacy of skin disinfectants against *C. acnes* and *S. aureus* dual-species biofilms. Reduction of *C. acnes* and *S. aureus* bacterial load in dual-species biofilms established in the presence of a combination of sebum-like components (i.e., SL) compared to reductions observed in mono-species biofilms following treatment with SD (2% chlorhexidine, 70% isopropyl alcohol), CP (Clinisept+) or SD and CP combined. Comparison in dual-species model consisting of (**A**) *C. acnes* isolate BPNBT-19269 and *S. aureus*, (**B**) *C. acnes* isolate BPNBT-19195 and *S. aureus*, (**C**) *C. acnes* isolate BPNBT-19329 and *S. aureus,* and (**D**) *C. acnes* isolate BPNBT-19227 and *S. aureus n* = 3, Welch’s *t*-test, *p* ≤ 0.05 (*), and *p* ≤ 0.01 (**).

**Figure 7 microorganisms-12-00271-f007:**
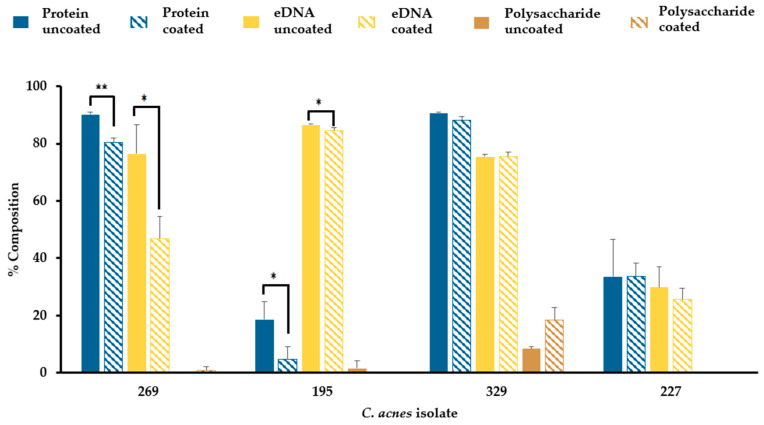
Composition of *C. acnes* biofilm extracellular matrix. Comparison of the extracellular matrix composition of *C. acnes* biofilms established in the absence or presence of a combination of sebum-like components (i.e., SL) as assessed by crystal violet staining following matrix disruption. *n* = 3, Welch’s *t*-test, *p* ≤ 0.05 (*), and *p* ≤ 0.01 (**).

**Table 1 microorganisms-12-00271-t001:** Bacterial identifiers of *C. acnes* isolates derived from contaminated PCs and the geographical region of component production.

*C. acnes* Isolate	Geographical Region (Canadian Province)
BPNBT-18005	Alberta
BPNBT-18120	Alberta
BPNBT-19210	Alberta
BPNBT-19223	Alberta
BPNBT-19269	Alberta
BPNBT-19329	Alberta
BPNBT-18313	British Columbia/Yukon
BPNBT-19119	British Columbia/Yukon
BPNBT-19195	British Columbia/Yukon
BPNBT-19280	British Columbia/Yukon
BPNBT-19054	Manitoba
BPNBT-19356	Manitoba
BPNBT-18290	Ontario
BPNBT-18317	Ontario
BPNBT-19227	Ontario
BPNBT-19295	Ontario
BPNBT-19352	Ontario
BPNBT-19354	Ontario
BPNBT-19217	Ontario
BPNBT-19355	Ontario

**Table 2 microorganisms-12-00271-t002:** Categorization of the 20 *C. acnes* PC isolates based on phylotyping touchdown-multiplex PCR [22].

*C. acnes* ID	Phylotype	% (*n* = 20)
BPNBT-19269	IA	5
BPNBT-19119, 19195, 19210, 19313, 19352, and 19356	IB	30
BPNBT-18290, 18317, 19005, 19054, 19120, 19217, 19280, 19295, 19329, and 19354	II	50
BPNBT-19223, 19227, and 19355	III	15

## Data Availability

The data of all results in this study are included in the manuscript. Raw data will be provided by the corresponding author upon reasonable request.

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
