# Peer review of "Sebum Components Dampen the Efficacy of Skin Disinfectants against Cutibacterium acnes Biofilms"

_microorganisms, 2024, doi:10.3390/microorganisms12020271_

Round 1

Reviewer 1 Report

Comments and Suggestions for Authors

This study, titled "Sebum Components Dampen the Efficacy of Skin Disinfectants against Cutibacterium acnes Biofilms," explores the impact of sebum components on the effectiveness of skin disinfectants against Cutibacterium acnes, a common contaminant in platelet concentrates. The main findings and methods include researching how sebum components affect the formation of C. acnes biofilms in both mono-species and co-cultured models with Staphylococcus aureus. Despite the use of standard skin disinfectants, C. acnes remains a major contaminant in platelet concentrates due to its ability to form biofilms in sebaceous environments.

I believe the article provides a detailed analysis and summary of how sebum components significantly weaken the efficacy of standard disinfectants against C. acnes biofilms. The study assessed 20 C. acnes isolates collected from various regions in Canada (2018-2019), along with a S. aureus isolate for dual biofilm experiments. Preliminary assessments of biofilm formation, including oleic acid supplementation and staining procedures, were conducted. The study also tested the impact of individual sebum components and their combinations on biofilm formation, followed by confocal microscopy analysis.

Disinfectant efficacy on both mono-species and dual-species biofilms was evaluated. Biofilm matrix disruption experiments were conducted to understand the composition of the extracellular matrix. This highlights the challenges of effectively disinfecting donor skin before blood collection.

However, I think there are areas in the article that need improvement, such as:

  1. 1.Why Figure 3A lacks a scale part.
  2. 2.The extracellular matrix of C. acnes biofilms, mainly composed of protein and extracellular DNA, is impacted by sebum components, potentially altering biofilm properties and resistance to disinfectants.

Regarding sebum components and the pathways related to damage response or sebaceous gland components, additional references should be included in the article (Veniaminova, N.A. et al. Cell Rep, 2023) (Han, J. et al. iScience, 2023) to better integrate into your theme: Staphylococcus aureus can form biofilms in the presence of pre-formed C. acnes biofilms. The bacterial concentrations were significantly reduced when co-cultured with certain C. acnes phylotypes. The study emphasizes the need for biofilm models that more accurately reflect the natural habitat of bacteria to better understand their response to antimicrobials and disinfectants.

Comments on the Quality of English Language

Minor editing of English language required

Author Response

1.Why Figure 3A lacks a scale part

ANSWER: Thank you for the observation. The scale was present in the figure but not visible. A new Scale has been added to the first image and it and applies to all images presented in Figure 3A.

2.The extracellular matrix of C. acnes biofilms, mainly composed of protein and extracellular DNA, is impacted by sebum components, potentially altering biofilm properties and resistance to disinfectants. Regarding sebum components and the pathways related to damage response or sebaceous gland components, additional references should be included in the article (Veniaminova, N.A. et al. Cell Rep, 2023) (Han, J. et al. iScience, 2023) to better integrate into your theme: Staphylococcus aureus can form biofilms in the presence of pre-formed C. acnes biofilms. The bacterial concentrations were significantly reduced when co-cultured with certain C. acnes phylotypes. The study emphasizes the need for biofilm models that more accurately reflect the natural habitat of bacteria to better understand their response to antimicrobials and disinfectants.

ANSWER: Our study relates specifically to the impact of skin sebum on disinfectant efficacy due to the structure of the bacterial biofilm matrix. We were only able to find an article from Veniaminova in Cell Reports related to direct mechanisms for sebaceous gland cell-renewal (https://www.cell.com/cell-reports/pdf/S2211-1247(23)01133-6.pdf). This work although interesting does not relate to the work presented in our study. The article by Han et al was not found without the full citation. We have therefore not included any of these references in the revised manuscript.

Reviewer 2 Report

Comments and Suggestions for Authors

this article is interesting since  it show that desinfectant efficacy is reduced in the presence of sebum components against all C. acnes biofilms tested  contributing to its over representation as a platelet concentrate contaminant. we miss a solution to decrease this phenomena as a cleanser wash with water prior to desinfection to limit sebum presence and destabilised biofilm. This type of possible choice might be include in the discussion of the article to give suggestion to doctor. 

Author Response

this article is interesting since  it show that desinfectant efficacy is reduced in the presence of sebum components against all C. acnes biofilms tested  contributing to its over representation as a platelet concentrate contaminant. we miss a solution to decrease this phenomena as a cleanser wash with water prior to desinfection to limit sebum presence and destabilised biofilm. This type of possible choice might be include in the discussion of the article to give suggestion to doctor.

ANSWER: We appreciate the comment; however, sebum is part of the deeper layers of skin, it would not be removed with water. The current skin blood donor skin disinfectant comprises chlorhexidine and isopropyl alcohol to increase penetration of the disinfectant. Unfortunately, as demonstrated in our work, the standard disinfectant is not efficient for C. acnes biofilms.

Reviewer 3 Report

Comments and Suggestions for Authors

This is a very promising study. I am impressed by the approach presented by the authors. After reading the manuscript, minor concerns were identified, which are the following:

1) The abstract is confusing and difficult to follow; please improve it by presenting clearly the objectives and main findings;

2) Introduction - This section is adequate and presents a suitable background to support the hypothesis of this study;

3) Material and methods - Is it necessary to have the approval of an Ethical committee to perform such a study? The authors obtained the samples from donors, which implies that humans are involved. Please clarify this matter.

4) Results - this section is adequate;

5) Discussion - The authors could include a brief paragraph approaching the main limitations of their study, aiming for a better data interpretation. Is it the first study that approached the matter? It should be reinforced. Despite the intriguing results, I felt that the main findings were not properly presented; The authors could summarize in the first paragraph of the discussion the findings and their meaning;

Can any interventions be applied to solve this matter based on the results?

A graphical abstract would positively impact the study.

Author Response

This is a very promising study. I am impressed by the approach presented by the authors. After reading the manuscript, minor concerns were identified, which are the following:

  • The abstract is confusing and difficult to follow; please improve it by presenting clearly the objectives and main findings;

ANSWER: We thank the Reviewer for the comment and the abstract has been re-written to improve flow of ideas.

2) Introduction - This section is adequate and presents a suitable background to support the hypothesis of this study;

3) Material and methods - Is it necessary to have the approval of an Ethical committee to perform such a study? The authors obtained the samples from donors, which implies that humans are involved. Please clarify this matter.

ANSWER: The bacterial isolates were obtained from contaminated platelet units screened as part of routine testing at Canadian Blood Services. No human voluntaries were recruited specifically for the study and no human samples were used during testing described in the manuscript. Ethical approval does not apply.

4) Results - this section is adequate;

5) Discussion - The authors could include a brief paragraph approaching the main limitations of their study, aiming for a better data interpretation. Is it the first study that approached the matter? It should be reinforced. Despite the intriguing results, I felt that the main findings were not properly presented; The authors could summarize in the first paragraph of the discussion the findings and their meaning;

ANSWER: We thank the Reviewer for this observation and the Discussion has been modified accordingly. A  statement summarizing the major findings has been moved to the first paragraph and a paragraph outlining limitations has been added to the end of the Discussion in the revised manuscript.

Can any interventions be applied to solve this matter based on the results?

ANSWER: Currently, no approved disinfectants can penetrate deep enough or overcome the inhibitory effects of the standard disinfectant. However, having the knowledge obtained in our study allows for the innovation of new disinfectants that are less susceptible to sebum inhibitory properties as included in the revised Conclusion of the manuscript.

A graphical abstract would positively impact the study.

ANSWER: We are grateful by the suggestion but as the abstract has been re-written based on the Reviewers’ comments, we consider that a graphical abstract would not add value to the revised manuscript.

Reviewer 4 Report

Comments and Suggestions for Authors

The aim of the present manuscript was to evaluate the effect of sebum components on disinfectant efficacy as assessed by in vitro assays against four PC C. acnes isolates in a monospecies and in a two-species biofilm model including a transfusion relevant Staphylococcus aureus isolate. The authors performed a series of in vitro experiments to test a new decontamination protocol. The manuscript presents the following issues: 

- The authors must define CP.

- Item 2.4. - Why coat the plates used to grow the biofilm with olive oil? This information should be included in the Introduction section.

- Confocal microscopy: Why fix with 2.5% glutaraldehyde? The advantage of working with confocal microscopy is to use live cells and the authors used dead cells?

- Disinfectant assays should have a positive control such as chlorhexidine or other well-established antimicrobial agent.

- The results of Figure 1 and 3B lack statistical analysis.

- Instead of using Crystal Violet for dual-species biofilm assessment, the authors should use XTT. Crystal violet marks dead bacteria, which may affect the result.

- The conclusion section should not include citations and references. Please shorten your conclusion and limit it to the conclusion of your experiments.

Author Response

The aim of the present manuscript was to evaluate the effect of sebum components on disinfectant efficacy as assessed by in vitro assays against four PC C. acnes isolates in a monospecies and in a two-species biofilm model including a transfusion relevant Staphylococcus aureus isolate. The authors performed a series of in vitro experiments to test a new decontamination protocol. The manuscript presents the following issues: 

- The authors must define CP.

ANSWER: CP is defined. Please see page 3, line 111 under 2.2 Media and reagents

- Item 2.4. - Why coat the plates used to grow the biofilm with olive oil? This information should be included in the Introduction section.

ANSWER: We believe the Reviewer is referring to item 2.3 Preliminary assessment of C. acnes biofilm formation with oleic acid supplementation.  The initial experiments with olive oil demonstrated a reliable method to improve bacterial attachment for biofilm formation, avoiding technical differences, and obtaining reliable/reproducible data. Olive oil demonstrated that biofilms were more stable and more difficult to dislodge during washes. Therefore, the same principle was applied for other sebum components (item 2.4 Impact of sebum components on C. acnes biofilm formation). We think that these items belong to Materials and Methods and not to introduction.

- Confocal microscopy: Why fix with 2.5% glutaraldehyde? The advantage of working with confocal microscopy is to use live cells and the authors used dead cells?

ASWER: Confocal microscopy was performed to observe overall topographical features of the biofilms. Specifically for our study, viability of the cells would not have added value to the features of the biofilm structure that we wanted to elucidate.

- Disinfectant assays should have a positive control such as chlorhexidine or other well-established antimicrobial agent.

ANSWER: We did include a positive control, it is the gold standard disinfectant used for blood donors, which is composed of chlorhexidine (mentioned by the Reviewer) and isopropyl alcohol, please see definition of the standard disinfectant (SD) in page 3, lines 109-110 of the manuscript. Then, the description of how the experiments were set up with the proper control (SD disinfectant) is found on page 5, section 2.8 Disinfectant efficacy assays.

- The results of Figure 1 and 3B lack statistical analysis.

ASWER: Thank you for this observation. Statistical analyses were performed and included in Figures 1 and 3B.

- Instead of using Crystal Violet for dual-species biofilm assessment, the authors should use XTT. Crystal violet marks dead bacteria, which may affect the result.

ANSWER: In addition to crystal violet (Figure 4A), viable counts were performed which provides viability data on each species in our model (Figures 4B and 4C).  An XTT assay would not allow to differentiate between the viability of the two species as done in our experimental design.

- The conclusion section should not include citations and references. Please shorten your conclusion and limit it to the conclusion of your experiments.

ANSWER: We are grateful for this comment and the revised conclusion has been shortened and citations have been removed.

Round 2

Reviewer 1 Report

Comments and Suggestions for Authors

I think it can be published.

Comments on the Quality of English Language

English is enough to comprehense.

Author Response

Thanks for your review

Reviewer 3 Report

Comments and Suggestions for Authors

The author's revision improved the document, and I suggest accepting it.

Author Response

Thanks for your review

Reviewer 4 Report

Comments and Suggestions for Authors

- I work with confocal analysis and the Thecniquine does not need 2.5% glutaraldehyde. Perhaps there is a confusion of names and the authors used a different microscopy technique. Maybe it is transmission electron microscopy?

- Crystal violet/viability test: If the authors want to differentiate the two species within the biofilm, a PCR assay would be much better. 

Author Response

- I work with confocal analysis and the Thecniquine does not need 2.5% glutaraldehyde. Perhaps there is a confusion of names and the authors used a different microscopy technique. Maybe it is transmission electron microscopy?

 ANSWER: The technique that was used to image the biofilm samples was confocal microscopy, and 2.5% glutaraldehyde was only used to fix the samples as the Facility where the images were taken do not allow live microbial cells. Glutaraldehyde was used as an alternative fixative to paraformaldehyde as per consultation with the technician who performed the imaging. This fixative has been successfully used for confocal microscopy of mitochondria (The Combination of Paraformaldehyde and Glutaraldehyde Is a Potential Fixative for Mitochondria - PMC (nih.gov)).

- Crystal violet/viability test: If the authors want to differentiate the two species within the biofilm, a PCR assay would be much better. 

ANSWER: We agree that a PCR-based assay could be an alternative method to differentiate species within a polymicrobial biofilm (if it is optimized to detect viable cells), but isolating and quantifying colonies from two species is a definitive method to achieve the same goal, and it has been optimized and used in our lab for previous studies.